# Identification and Characterization of a Novel BVDV-1b Cluster in Sardinia Through Whole Genome Sequencing

**DOI:** 10.3390/v17121606

**Published:** 2025-12-12

**Authors:** Giada Lostia, Elisabetta Coradduzza, Loris Bertoldi, Angela Maria Rocchigiani, Roberto Bechere, Cinzia Pasini, Lorenzo Stevanato, Mariangela Stefania Fiori, Angelo Ruiu, Giantonella Puggioni

**Affiliations:** 1Istituto Zooprofilattico Sperimentale della Sardegna, 07100 Sassari, Italy; giada.lostia@izs-sardegna.it (G.L.); angelamaria.rocchigiani@izs-sardegna.it (A.M.R.); roberto.bechere@izs-sardegna.it (R.B.); veterpasini@gmail.com (C.P.); lorenzo.stevanato@izs-sardegna.it (L.S.); mariangela.fiori@izs-sardegna.it (M.S.F.); angelo.ruiu@izs-sardegna.it (A.R.); giantonella.puggioni@izs-sardegna.it (G.P.); 2BMR Genomics s.r.l., 35131 Padova, Italy; loris.bertoldi@bmr-genomics.it

**Keywords:** bovine diarrhea virus (BVDV), Whole Genome Sequencing (WGS), phylogenetic analysis, variant analysis

## Abstract

Bovine viral diarrhea (BVD) is a highly infectious disease with a global distribution caused by the bovine viral diarrhea virus (BVDV), primarily affecting cattle. Dairy farms play a central role in the persistence and spread of BVDV in Italy, making control strategies and genetic studies essential to reduce its circulation. This work aimed to identify and characterize the genotype and subgenotype of BVDV infecting cattle in a specific area of Sardinia. Ten BVDV Sardinian strains were isolated and sequenced from the blood of infected cattle collected into EDTA tubes during outbreaks between 2018 and 2024. Then, to characterize the isolates, phylogenetic and variant analyses were performed on the entire collection of BVDV genomes available to date in GenBank. All Sardinian isolates were assigned to the BVDV-1b subgenotype. Except for two divergent strains, the isolates clustered into a distinct monophyletic clade characterized by 61 exclusive variants absent in all other analyzed sequences. These findings point to the existence of a distinct Sardinian genomic signature. Moreover, among these mutations, 19 missenses distributed on genes encoding the E1, E2, Core, NS3, NS4B and NS5A proteins could have a relevant functional impact, given the role these proteins play in the virus life cycle and in interaction with the host.

## 1. Introduction

Bovine viral diarrhea (BVD) is a highly infectious disease caused by the Bovine viral diarrhea virus (BVDV), a globally distributed pathogen belonging to the *Pestivirus* genus of the Flaviviridae family [1,2].

BVDV is a single-stranded, positive-sense RNA virus with a genome consisting of approximately 12.3 kb in length. The viral genome features a single open reading frame encoding a polyprotein composed of 3900 amino acids, which is post-translationally cleaved into four structural (C, E^rns^, E1, and E2) and eight non-structural (N^pro^, P7, NS2-3 or NS2, NS3, NS4A, NS4B, NS5A, and NS5B) proteins [3], flanked by a 5′ and 3′untranslated regions (5′-UTR and 3′-UTR) of about 360–390 and 200–240 nucleotides, respectively [4,5]. Currently, the genus is classified into 11 species designated Pestivirus A to Pestivirus K, based on genetic variability, antigenic differences and natural host range [6]. BVDV is assigned into three species: Pestivirus A or BVDV-1 (or *Pestivirusbovis*), Pestivirus B or BVDV-2 (or *Pestivirustauri*), and Pestivirus H or BVDV-3 (*Pestivirusbrazilense* or *HoBi-like* virus) [6,7].

To date, due to its genetic heterogeneity, BVDV species have been divided into subtypes based on partial and complete genome sequences: BVDV-1 into 23 (from a to x); BVDV-2 into 5 (from 2a to 2e); and BVDV-3 into 5 (from a to e) [2,7,8,9,10,11]. Furthermore, BVDV isolates are classified into two distinct biotypes based on their effect on cultured cells: cytopathic (cp), which induces cell death, and noncytopathic (ncp), which replicates without causing visible cytopathic effects (CPE) or compromising cell viability [12,13]. The ncp types are more prevalent and often associated with severe acute infections, while the cp types are less common and usually involved in outbreaks of Mucosal Disease [8,13] in persistently infected (PI) animals (animals born from infected pregnant females) [14].

The natural hosts of BVDV-1, BVDV-2, and BVDV-3 have been identified as cattle of all breeds and ages; however, these genotypes are also capable of infecting 50 other ruminants and clovenhoofed animal species (e.g., sheep, goats, pigs, giraffes, camels) [2,3,15].

Infection with BVDV induces viral diarrhea characterized by a wide spectrum of clinical manifestations from fever, pyrexia, and leukopenia to weakness, poor growth, immunosuppression, abortion, fetal death, delivery of PI animals, reduced milk production, and an increasing incidence of other diseases [3].

Collectively, these factors exert a substantial impact on both animal welfare and the economic sustainability of dairy and beef cattle sectors [16].

The transmission of BVDV can occur in many ways, including direct contact, through various excreta (urine, feces, excrement, milk, and semen), contaminated materials, and vertical transmission, which plays a significant role in its epidemiology and pathogenesis [17]. Indeed, PI animals make it difficult to manage and control BVDV transmission: their immune tolerance to infectious strains, the absence of antibody production, and the persistent infection, lead them to carry and spread BVDV throughout their lives [17].

To identify BVDV subgenotypes, elucidate their evolutionary history, and determine their prevalence geographically, phylogenetic studies have primarily relied on the analysis of the highly conserved 5′-UTR, followed by N^pro^ and E2 coding sequences [18,19].

To date, these studies have demonstrated that BVDV-1 has a more widespread distribution than BVDV-2 and BVDV-3 viruses [18]. In detail, within BVDV-1, it has been reported that the BVDV-1b subgenotype exhibits a global predominance, followed by BVDV-1a and BVDV-1c; regarding BVDV-2, subgenotype 2a has been identified as the most prevalent worldwide, while subgenotypes 2b, 2c, and 2d have been detected exclusively in European and Asian countries; finally, BVDV-3 subgenotypes have been detected only in Europe, Asia, Africa (only Egypt) and South America [18,19,20].

In Italy, Pestivirus infection has been reported in cattle throughout the country since 1960 and has been recognized as a relevant disease in cattle herds since 1990 [21].

To date, studies have revealed a wide BVDV genetic heterogeneity: sequence analysis divided the BVDV isolates into BVDV-1, BVDV-2, and BVDV-3 (detected only in 2010) [22,23], and at subgroup level into BVDV-1a, BVDV-1b, BVDV-1c, BVDV-1d, BVDV-1e, BVDV-1f, BVDV-1g, BVDV-1h, BVDV-1j, BVDV-1k, BVDV-1l, BVDV-1r, BVDV-1s, BVDV-1t, BVDV-1u, BVDV-2a and BVDV-2c [21,24,25,26,27,28,29]. Regarding frequency, BVDV-1 has been identified as the most prevalent, with 96.9% of the strains detected, while BVDV-2 and BVDV-3 have lower and sporadic rates of circulation, respectively [21].

The introduction, the spatio-temporal distribution and the genetic heterogeneity of the outbreaks of BVDV suggest that this could mainly be attributed to livestock trade flows (including cattle trade and movements) within the country, or to the use of contaminated biological products (e.g., contaminated by fetal bovine serum), or to the introduction of virus strains from other countries due to the lack of systematic control measures [21]. In addition, the high level of genetic diversity present in Italy may have implications for diagnostic and immunology by affecting the performance of diagnostic tools and the cross-protection of commercially available vaccines [21]. Specifically, in Italy, dairy farms have been identified as a primary source of persistence and dispersion of BVDV [30]. Therefore, the implementation of control measures targeted at these herds would lead to a substantial reduction in BVDV circulation in Italian cattle.

Therefore, this study aimed to characterize the BVDV genotypes and subgenotypes circulating in a high-density dairy area of Sardinia. By establishing a systematic epidemiological framework, this work seeks to guide targeted intervention strategies and refined diagnostic tools applicable to other at-risk regions.

## 2. Materials and Methods

### 2.1. Sampling

The 10 pathological samples included in this study were collected between 2018 and 2024 from naturally infected cattle during different outbreaks occurring in the municipality of Arborea, in the province of Oristano (Figure 1).

### 2.2. Viral RNA Extraction and Sequencing Independent Single Primer Amplification (SISPA) Protocol

Viral RNA was extracted from organs (kidneys, brain, intestine) or blood in EDTA samples using the Qiagen Viral RNA Mini Kit (Qiagen, Hilden, Germany) according to the manufacturer’s instructions.

The quantification of the extracted RNA was then conducted using the Qubit™ RNA High Sensitivity kit on the Qubit 4 Fluorometer (Invitrogen, Thermo Fisher Scientific, Waltham, MA, USA), as described in the user’s manual instructions.

A total of 60 ng of RNA was used for the assessment of the SISPA as described by Marcacci et al. 2016 [31] with one modification. Specifically, due to the reduced volume of sample eluates, the RNA digestion with DNAse prior to quantification was not performed. RNA was reverse-retrotranscribed into single strand cDNA (ss cDNA) using SuperScript^®^ IV Reverse Transcriptase (200 U, Thermo Fisher Scientific, Waltham, MA, USA) with the random primer FR26RV-N (50 µM), SSIV buffer (1X), dNTPs (10 mM), DTT (100 mM) and RNase Recombinant Ribonuclease inhibitor (RNAseOUT, 40 U, Thermo Fisher Scientific, Waltham, MA, USA).

The reaction was incubated at 23 °C for 10 min and 50 °C for 50 min. After an inactivation step at 80 °C for 10°, 1 µL of Klenow Fragment (3′→5′ exo-) (5 U/µL, New England Biolabs, Ipswich, MA, USA) was added directly to the reaction tube to perform second-strand cDNA (ds cDNA) synthesis, and the incubation was carried out at 37 °C for 1 h and 75 °C for 10 min. Finally, 5 µL of the obtained ds cDNA was added to the PCR master mix containing 25 µL of PfuUltra II Hotstart 2X Master Mix (Agilent Technologies, Santa Clara, CA, USA), 1 µL of primer-tag FR20RV (40 µM), and 19 µL of RNase-free H_2_O. Incubation was performed at the following thermal conditions: 95 °C for 1 min, 40 cycles of 95 °C for 20 s, 65 °C for 20 s and 72 °C for 2 min, and a final extension step of 72 °C for 3 min.

The PCR product was then purified to remove all traces of the components of the enzymatic reaction using the GeneAll^®^ Expin™ CleanUp SV kit (GeneAll Biotechnology Co., Ltd., Seoul, Republic of Korea) and quantified using the Qubit™ DNA HS assay kit (Invitrogen, Thermo Fisher Scientific, Waltham, MA, USA).on Qubit 4 Fluorometer (Invitrogen, Thermo Fisher Scientific, Waltham, MA, USA).

### 2.3. Sequencing and Genome Assembly

Sequencing was performed on ten BVDV Sardinian strains (PV871979, PV871980, PV871981, PV871982, PV871983, PV871984, PV871985, PV871986, PV871987, PV871988) at BMR Genomics s.r.l. (Padova, Italy). Sequencing libraries were prepared using the Illumina DNA Prep kit (Illumina, San Diego, CA, USA) according to the manufacturer’s protocol. Library quality and fragment size distribution were assessed using an Agilent Bioanalyzer (Agilent Technologies, Santa Clara, CA, USA), and quantification was performed using a Qubit fluorometer (Thermo Fisher Scientific, Bedford, MA, USA).Finally, they were loaded into the Illumina MiSeq (PV871981, PV871982, PV871983, and PV871986) or Illumina NovaSeq (PV871979, PV871980, PV871984, PV871985, PV871987, and PV871988) sequencers (Illumina, Inc., Ann Arbor, MI, USA) and sequenced according to the V3-300PE or 150PE strategy, respectively, as summarized in Table 1. NovaSeq sequencing was used only for samples that performed poorly in terms of viral reads with MiSeq.

Initial read quality assessment was performed using FastQC (v0.11.9). Subsequently, reads were preprocessed with fastp (v0.24.0) [32] to remove residual primer sequences, low-quality (--qualified_quality_phred 20, --unqualified_percent_limit 30 and --average_qual 25) and low-complexity (--low_complexity_filter and --complexity_threshold 30) reads, and those shorter than 50% of the read length (150 and 75 for 300PE and 150PE, respectively). The remaining reads were assembled using Geneious Prime (GP) (2025.1.1) [33], guided by the BVDV Osloss strain genome (GenBank accession M96687.1), applying a medium-low sensitivity protocol. Assembly quality was assessed using QUAST (v5.3.0) [34]. Gene prediction was performed with prokka (1.14.6) [35] (kingdom: Virus, genetic code 1), and gene annotation was carried out using eggNOG-mapper (v2.1.12) [36].

A total of 236 nucleotide sequences, including a selection of BVDV genomes, the 10 reference-guided assembled sequences of the Sardinian strains, and two outgroup genomes U70263 and EU497410, corresponding, respectively, to the Border disease and swine fever viruses, were used to perform a multiple sequences alignment (MSA) by using MAFFT (v7.505) [37]. The best-fit nucleotide substitution model was identified using ModelTest-NG (v0.1.7) [38]. The analysis was configured for nucleotide sequences, utilizing specific heuristic search settings and empirical base frequencies (parameters: -d nt -h uigfr -f ef). The evolutionary analysis was conducted in MEGA X (v11.0.13) [39] employing the Maximum Likelihood method based on the General Time Reversible model [40] with gamma-distributed rate variation (GTR+G; parameter = 0.8471). Heuristic searches utilized initial trees obtained via Neighbor-Joining and BioNJ algorithms applied to Maximum Composite Likelihood distance matrices. The final dataset was filtered using a partial deletion threshold of 95% for site coverage, and nodal support was evaluated through 100 bootstrap replicates. The resulting phylogenetic tree was built using iTOL (v7) [41].

The same-236 genomic sequences were further analyzed using snippy-multi (v4.6.0) [42] with the following parameters (--mapqual 30 --mincov 10 --minqual 40 --subsample 1 --fbopt “--haplotype-length −1”), to characterize variants (SNP, del, ins) differentiating the various strains from the main BVDV-1b genome (KY964311).Coverage analysis was performed using mosdepth (v0.3.10) [43] with standard parameters on a BVDV-1b subset of samples, including all 10 Sardinian strains (n_tot_ = 52). Custom Python scripts were designed to process snippy-multi results in order to determine variants characterizing Sardinian samples and their distribution in the other analyzed samples, and to process coverage results (Phyton 3.13.0). Functional impact prediction was performed on a selected subset of polymorphisms using the SIFT web server [44]. The UniProt/SwissProt database was used as the reference to determine whether the resulting amino acid changes were likely to compromise protein function (ATN39078.1 polyprotein of KY964311).

## 3. Results

### 3.1. Reads Preprocessing

The whole genome sequencing (WGS) of the ten BVDV strains under investigation (designated with the following identifiers: PV871979, PV871980, PV871981, PV871982, PV871983, PV871984, PV871985, PV871986, PV871987, PV871988) was performed using MiSeq and NovaSeq Illumina sequencing platforms and yielded 114,836,246, 106,501,370, 2,233,016, 2,067,572, 1,741,588, 144,038,856, 103,669,536, 1,383,686, 122,942,354, 163,000,356 bp paired-end (PE) reads, respectively. Although all samples produced a great and satisfactory number of reads, most were removed during the pre-processing stage, as shown in Table 2.

The vast majority of discarded reads originated from host contamination, a direct result of not using viral enrichment steps prior to sequencing. Given the low viral-to-host ratio, a reference-guided assembly strategy was adopted as the most effective approach. While this method inherently risks masking highly divergent or novel genomic regions, it significantly enhanced overall assembly integrity by minimizing the frequency of misassembled contigs typical of high-background datasets.

### 3.2. Genome Assembly

The assembly process for the cleaned reads using Geneious Prime (GP) was guided by the Osloss genome (M96687.1) rather than the reference genome of BVDV (NC001461_1). This approach allowed obtaining better results, producing a lower number of unknown bases (N) and a higher completeness in the final sequences. Information on the ten assembled genomes using GP is collected in Table 3. Although the number of ambiguous bases is not negligible, inspection of the assemblies confirmed that these gaps were mainly restricted to the extreme 5′ and 3′ termini, leaving the complete Open Reading Frame (ORF) intact for phylogenetic and molecular characterization.

### 3.3. Phylogenetic Analysis

The evolutionary history of BVDV was inferred using the Maximum Likelihood method and the General Time Reversible (GTR) model. The optimal tree with the highest log likelihood value (InL = −450824.70) is presented in Figure 2. Nodal support values were calculated from 100 bootstrap replicates. This analysis was performed using MEGA11 and involved 236 whole-genome sequences (234 BVDV, 1 BDV, and 1 SFV-1) with 11,853 analyzed positions in the final dataset. This high coverage (>94% of the genome length) ensured that the phylogenetic inference was based on comprehensive genome-wide signal rather than limited genomic regions. The Sum of Branch Length (SBL) is 8.84128044, while the ratio Ts/Tv is 4.4947. The robustness of the phylogenetic inference was further assessed by testing different alignment filtering parameters, including a stricter 99% partial deletion threshold and the retention of the full alignment (>23,900 positions). In all simulated scenarios, the tree topology remained consistent, confirming the stability of the analysis.

Phylogenetic analysis demonstrated that the majority of Sardinian isolates clustered together, forming a distinct, highly supported monophyletic clade (bootstrap = 100%). Exceptions were observed for strains PV871986 and PV871987, which displayed divergent clustering patterns. The former formed a clade with strain EF101530 (DEU1) with 100% bootstrap support, whereas PV871987 was positioned within a broader cluster characterized by moderate nodal support (68%). Notably, all Sardinian sequences were assigned to the BVDV-1b subtype, falling within a larger clade comprising 52 strains with maximum bootstrap support (100%).

### 3.4. Variant Analysis

The results obtained from the phylogenetic analysis were then used to perform a variant analysis to characterize variations detectable in the Sardinian strains using KY964311 (BVDV 1b) as a reference.

The analysis revealed that despite the Sardinian strains’ proximity to the reference genome within the phylogenetic tree, they differed from it by a high number of variations, ranging from 721 to 950 (Appendix A). However, the majority of them do not highlight an important impact at the protein level, as they are mostly synonymous variations (see Table 4, Figure 3).

When considering all 10 Sardinian samples together, it was observed that they did not exhibit any group-characterizing variants despite the fact 221 variants were shared among all samples. However, a different pattern emerged upon excluding the two most phylogenetically divergent samples (PV871986 and PV871987). The resulting subset, which includes samples PV871979, PV871980, PV871981, PV871982, PV871983, PV871984, PV871985, and PV871988, revealed 529 shared variants, of which 61 (as detailed in Appendix A) were found exclusively within them and not in the other analyzed samples. These 61 variants are mainly synonymous and missense and are distributed along the entire genome, with the exception of the portion coding for the P7 protein, and the last fragment represented by NS5B and 3′UTR (as shown in Table 5). Functional assessment using SIFT against the UniProt database suggested that this subset of the lineage-specific variants might compromise protein function. However, it must be noted that the majority of these predictions were associated with low confidence scores, likely due to the limited diversity of BVDV sequences in the reference database used by the tool.

To assess the robustness of the genomic signatures exclusive to the Sardinian cluster, we evaluated the base coverage at each variant position across the 52 BVDV-1b background genomes previously described. Variants were considered informative only if they met specific quality criteria: a minimum sequencing depth of 10× and a site completeness of at least 80% (i.e., the position was successfully covered in ≥42/52 background samples). This stringent filtering ensured that the identified polymorphisms reflected genuine evolutionary divergence rather than artifacts resulting from missing data in the analyzed sequences. Following this selection, 45 out of the 61 initially identified variants were retained. Collectively, these polymorphisms represent a promising set of molecular markers capable of discriminating the eight Sardinian isolates from other circulating BVDV-1b strains.

## 4. Discussion

Dairy farms play a crucial role in both the persistence and spread of the Bovine Viral Diarrhea Virus in Italy [30]. This highlights the importance of implementing targeted control measures of these herds to substantially reduce the spread of this virus among the Italian cattle. For this reason, research studies on the genetic and epidemiological characteristics of this virus are essential as they provide valuable information on the diversity of viral strains present in a specific area that could be helpful to develop intervention plans for the treatment and prevention of its circulation.

In this context, this work aimed to identify, characterize, and describe the genotypes and subgenotypes of BVDV circulating in a specific area in Sardinia with high-bovine-dairy production, providing information and enhancing our understanding of its epidemiological framework.

In this study, we characterized the genomes of ten BVDV strains obtained from EDTA-treated blood samples of naturally infected cattle in Sardinia during outbreaks between 2018 and 2024 using the Whole Genome Sequencing (WGS) approach. Inevitably, the application of WGS without prior specific viral enrichment resulted in a predominance of host reads, reflecting the inherent challenge of sequencing directly from complex clinical matrices. In order to overcome the consequent low viral-to-host ratio, we adopted a reference-guided assembly strategy. Although this method limits the detection of highly divergent genomic features compared to de novo approaches, it was essential to ensure high-quality consensus and avoid structural errors driven by the overwhelming background noise.

To determine the phylogenetic placement of the Sardinian strains, the assembled full-length genomes were compared against a comprehensive dataset of 224 BVDV whole-genome sequences available in NCBI GenBank. This genome-wide approach, in which 11,853 nucleotide positions were considered, provides a robust phylogenetic resolution, significantly reducing stochastic errors and increasing the reliability of the inferred topology. The reliability of this inferred topology was further corroborated by its stability across different alignment filtering stringencies, confirming that the high-resolution signal of the WGS data overcomes potential biases related to assembly gaps.

Phylogenetic analysis revealed that all sequenced isolates clustered within the BVDV-1b subtype. The high prevalence of BVDV-1b is of significant clinical relevance, as this subgenotype is frequently recovered from persistently infected animals and is a major contributor to BVDV-associated immunosuppression and secondary infections [47,48,49]. In addition, this finding aligns with the national epidemiological landscape described by Luzzago and Decaro [21], who identified BVDV-1b and BVDV-1e (69.5% to 2021) as the most prevalent and widely distributed subgenotypes in Italy, confirming their circulation across the entire country, including insular regions.

Most Sardinian isolates constituted a novel monophyletic group. However, strains PV871986 and PV871987 diverged from this main cluster, grouping instead with EF101530 (DEU1) and KF501393 (CHN 2009) sequences, respectively. The segregation of the primary Sardinian clade is also consistent with the unique genomic signatures identified by variant analysis. Notably, within the global dataset, the eight Sardinian strains described here form the largest and most recent monophyletic group of Italian BVDV-1b full genomes, highlighting the distinct and active circulation of this lineage.

Among all the observed variations (ranging from 721 to 950, predominantly synonymous) between the Sardinian strain and the phylogenetically proximate BVDV-1b reference (KY964311). Additionally, when considered altogether, the identified mutations did not exhibit any group-characterizing pattern.

However, a different interesting pattern emerged upon the exclusion of the two most phylogenetically divergent samples. In fact, the subset of the eight Sardinian strains shared 529 variants, of which 61 were found exclusively within them and not in any of the other analyzed samples, suggesting that they may represent a set of fully characterizing variants for Sardinian samples. Of these 61, 19 missense mutations were distributed along the genes encoding for the E1, E2, Core, N^pro^, NS2, NS3, NS4B, and NS5A proteins. Given the critical roles these proteins play in the viral life cycle and host interaction [50,51], the SIFT Sorting Intolerant from Tolerant (SIFT) algorithm was used to predict whether this amino acid substitutions could affect the function of these proteins [52]. The prediction indicated that among the 19 detected missense mutations, 7 are tolerated, 1 is unknown, and 11 affects the protein function. Specifically, the latter involved the Core (2), E1 (2), E2 (3), NS3 (1), NS4B (1), and NS5A (2) proteins. The Core and the three envelope glycoproteins E^rns^, E1, and E2, are the four Pestiviruses’ structural proteins [53].

The Core protein has been reported to exhibit both RNA chaperone and RNA-binding activities [54]. The analysis detected two missense mutations, c.686G>A p.Ser229Asn and c.691A>G p.Asn231Asp, and revealed a high degree of conservation, indicating that the Coreprotein stability is likely conserved within this cluster.

Of the three glycoproteins, the E1 protein’s function is the least well understood in terms of its functionality. Furthermore, the antigenic structure and epitopes of the protein have yet to be resolved [50]. However, the protein certainly is implicated in entering the host cell, interacting with other viral proteins during the entry process, and evading the immune response and virulence [50]. Of the two missense mutations detected on the protein (c.1629A>G p.Ile543Met; c.2041A>T p.Ile681Leu) in silico predictions via SIFT suggest that only the c.1629A>G p.Ile543Met substitution could potentially affect its function by modifying the interaction with the E2 protein.

The E2 protein is the most important of the three pestiviral glycoproteins because it is essential for host cell attachment, as it interacts with cell surface receptors that determine cell tropism, and for induce the antibody and cytotoxic T-lymphocyte responses neutralization [50]. Of the three missense mutations detected, only the c.2927T>C p.Ile976Thr and the c.2974A>T p.Thr992Ser, were identified as reliable markers. In fact, in silico predictions obtained via SIFT suggest that these substitutions could potentially affect protein function by changing the polarity of amino acids, resulting in potential alterations in viral tropism, modulation of cell entry efficiency, as well as facilitating immune evasion by modifying key antigenic epitopes.

Finally, 4 missense mutations were detected in the NS3, NS4A, and NS5B non-structural proteins, which could potentially alter their functionality. The NS3 protein is an essential component of the viral replicase, processing all downstream cleavage sites in synergy with its cofactor NS4A [51]. In relation to Pestiviruses, it has been established that NS3, NS4A, NS4B, NS5A, and NS5B coalesce into active viral RNA replicase in conjunction with an as-yet-unidentified number of host factors. The role of Pestivirus NS4B in virion morphogenesis remains to be fully elucidated; however, it has been demonstrated to be crucial for HCV virion morphogenesis, suggesting a similar function in Pestiviruses [51,55,56]. In silico predictions via SIFT suggest that these substitutions could potentially affect protein functions.

Collectively, it can be hypothesized that the accumulation of missense mutations in these key genes is likely to contribute to the genetic diversity observed in Sardinian strains and can significantly impact viral fitness, immune response evasion, and the resulting clinical manifestations of infection [50].

## 5. Conclusions

The results of this study improved our understanding and expanded our knowledge about the epidemiological framework of BVDV in Sardinian cattle, in particular, the characteristics and distribution of the strains responsible for the infection. Furthermore, it provided a genetic baseline for future studies, which are essential considering the high prevalence of BVDV infection. Indeed, a future objective includes ongoing monitoring of BVDV distribution in Sardinia, typing and analyzing the variants in a greater number of strains, in order to identify genetic mutations that could be useful in the development of tools such as vaccines [57], essential in the surveillance and control plans aimed at limiting its spread.

## Figures and Tables

**Figure 1 viruses-17-01606-f001:**
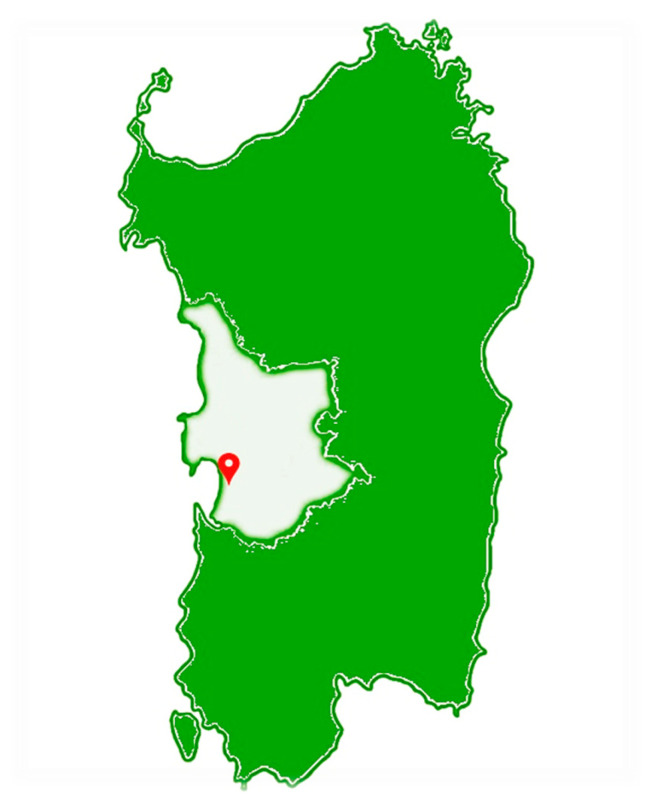
Map showing the location of the samples analyzed in the present study.

**Figure 2 viruses-17-01606-f002:**
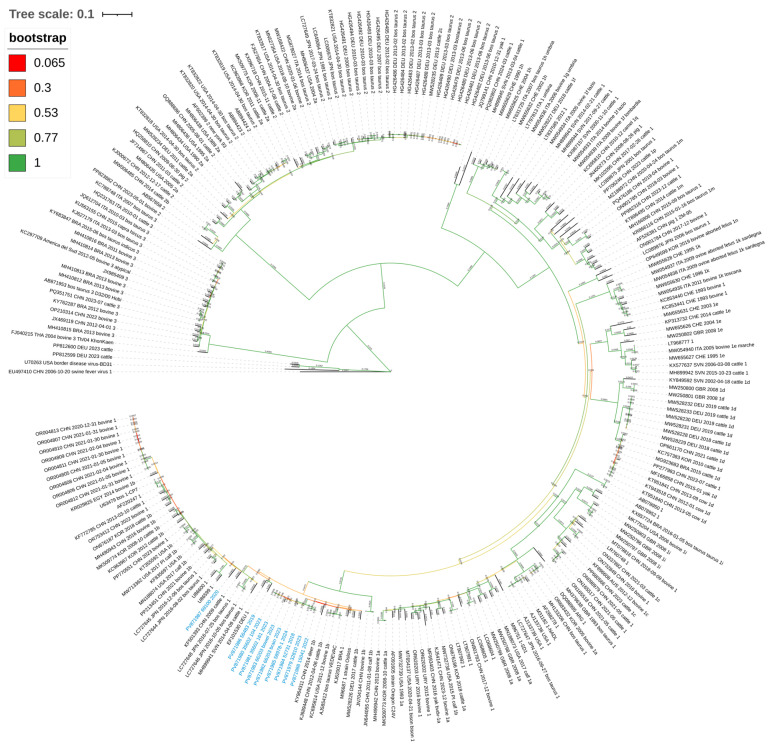
Evolutionary relationships of taxa. The figure shows the optimal unrooted tree for representing the evolutionary history of BVD viruses, inferred using the Maximum Likelihood method. The percentage of replicate trees in which the associated taxa clustered together in the bootstrap test (100 replicates) colored the branches, while the evolutionary distance is indicated above them. BDV and SFV-1 viruses are used as outgroups. Sardinian strains are highlighted in blue.

**Figure 3 viruses-17-01606-f003:**
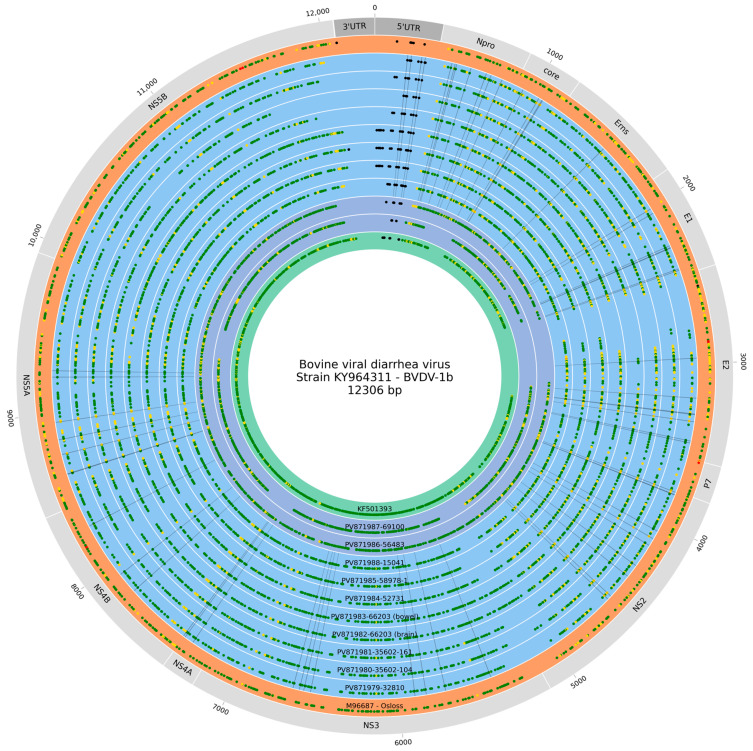
Sardinian strains variant analysis. Graphical representation of the variant analysis drawn with a customized version of Python library pyCircos [45]. The outer ring depicts the genomic structure of the BVDV-1b strain used as reference (KY964311). Inner circles show the distribution of variants (dots) in the analyzed strains along the reference genome. The background color of the rings distinguishes the different sample groups included in the analysis: the Osloss strain, utilized as the scaffold for genome assembly, is shown in orange; the highly similar Sardinian samples are in light blue; the two Sardinian outgroup samples are in violet, while their closely related strain KF501393 (CHN 2009) is represented in green. Dot color indicates the variant impact, assigned according to the work of Cingolani and colleagues [46], red for high, yellow for moderate, green for low, and black for modifier. The 61 variants characterizing the Sardinian strains are highlighted with a black background. Genome size is reported in bp in the external area.

**Table 1 viruses-17-01606-t001:** Sequencing information related to the analyzed samples.

Sample ID	PV871979	PV871980	PV871981	PV871982	PV871983	PV871984	PV871985	PV871986	PV871987	PV871988
Sequencing Platform	NovaSeq	NovaSeq	MiSeq	MiSeq	MiSeq	NovaSeq	NovaSeq	MiSeq	NovaSeq	NovaSeq
Sequencing Strategy	150PE	150PE	300PE	300PE	300PE	150PE	150PE	300PE	150PE	150PE

**Table 2 viruses-17-01606-t002:** Statistics on the number of reads through the various preprocessing steps. Read Number is calculated as the sum of the number of forward and reverse reads. High Quality reads correspond to fastp preprocessed reads. Assembled reads are those reads included within the assembled genomes. % Assembled corresponds to the fraction of assembled reads to the starting reads number.

Sample ID	PV871979	PV871980	PV871981	PV871982	PV871983	PV871984	PV871985	PV871986	PV871987	PV871988
Read Number	114,836,246	106,501,370	2,233,016	2,067,572	1,741,588	144,038,856	103,669,536	1,383,686	122,942,354	163,000,356
High Quality	108,538,404	101,522,160	1,762,376	1,659,102	1,159,380	119,946,540	95,299,274	540,206	107,756,610	152,216,554
Assembled	12,965	61,141	955	7731	16,914	15,855	83,855	3018	1468	6002
% Assembled	0.01217	0.05740	0.04277	0.37392	0.97118	0.01101	0.08089	0.21811	0.00119	0.00368

**Table 3 viruses-17-01606-t003:** Assembly metrics related to the Sardinian strains’ genomes, assembled with Geneious Prime. Ambiguous bases include both uncalled bases (N) and other IUPAC ambiguity codes representing multiple possible nucleotides (e.g., R, Y, S, W, K, M, B, D, H, V).

Sample ID	PV871979	PV871980	PV871981	PV871982	PV871983	PV871984	PV871985	PV871986	PV871987	PV871988
# Contigs	1	1	1	1	1	1	1	1	1	1
Total length (bp)	12,476	12,882	12,477	12,475	12,713	13,109	12,630	12,838	12,478	13,129
% GC	45.93	46.21	46.41	45.82	45.85	45.88	46.61	46.06	46.25	45.84
Coverage	163	1488	20	161	354	289	>5000	1736	17	157
# N	196	203	826	244	69	13	196	290	409	77
# Ambiguous bases	253	233	842	274	211	49	234	300	472	134
# Ambiguous bases per 100 kbp	2027.89	1808.73	6748.42	2196.39	1659.72	373.79	1852.73	2336.81	3782.66	1020.64

**Table 4 viruses-17-01606-t004:** Functional classification of variants identified in each isolate. The table reports the total number of variants categorized by their predicted molecular effect (e.g., synonymous, missense, intergenic, frameshift and stop gained).

Sample ID	PV871979	PV871980	PV871981	PV871982	PV871983	PV871984	PV871985	PV871986	PV871987	PV871988
Synonymous	585	591	553	591	647	586	600	735	773	607
Missense	162	171	162	149	180	172	163	182	169	174
Intergenic	7	8	6	8	14	16	13	9	7	11
Frameshift	0	0	0	0	0	0	0	0	0	1
Stop gained	0	0	0	0	0	0	0	0	1	0
Total	754	770	721	748	841	774	776	926	950	793

**Table 5 viruses-17-01606-t005:** The table shows the distribution of the 61 Sardinian strains’ characterizing variants along the BVDV-1b genome. The separation, depending on its impact at the protein level, is also reported.

Prod	5′UTR	N^pro^	core	E^rns^	E1	E2	P7	NS2	NS3	NS4A	NS4B	NS5A	NS5B	3′UTR	Total
Syn	0	6	1	1	5	4	0	4	7	2	3	5	0	0	38
Mis	0	1	3	0	2	3	0	6	1	0	1	2	0	0	19
Other	4	0	0	0	0	0	0	0	0	0	0	0	0	0	4
Total	4	7	4	1	7	7	0	10	8	2	4	7	0	0	61

## Data Availability

The sequences of the BVDV whole genomes obtained during the present study are openly available in the GenBank nucleotide sequence database under the accession numbers: PV871979, PV871980, PV871981, PV871982, PV871983, PV871984, PV871985, PV871986, PV871987, and PV871988.

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
