# Peer review of "Identification and Characterization of a Novel BVDV-1b Cluster in Sardinia Through Whole Genome Sequencing"

_viruses, 2025, doi:10.3390/v17121606_

Round 1
Reviewer 1 Report
Comments and Suggestions for Authors
This study provided important information of BVDV strains in Sardinian from 2018 to 2024. The topic falls within the scope of the journal, and the manuscript is well-organized and written. After a careful review, I recommend minor revisions.
1. Page1 line 41-42, for BVDV subtypes, the designations BVDV-1x and BVDV-2e have been tentatively proposed in several studies (e.g., BVDV-1x in Mucellini CI et al., Virus Genes 2023, 59(6):836–844; BVDV-2e in de Oliveira et al., Arch. Virol. 2022, 167:2545–2553). It is recommended that the authors update this information accordingly.
2. Page 2 line 69, for the description of BVDV proteins, the Erns("rns"is superscript) is commenly used than E0. for Npro, the "pro" should be superscript.
3. Page 2 line 83, The BVDV-3 was recently reported in dairy cattle herds in Egypt, Africa (Afify et al., Trop. Anim. Health Prod. 2022, 54, 336. ). This information can be updated.
4. Page7 line 229, are there any evidences the "BVDV 1b is particularly concerning because it is often associated with persistent infections and immunosuppression”?If so, it is better to list a reference or discuss this in the discussion part.
5. in the the discussion part, it is better to address the epidemiological links between the BVDV-1b viruses found in Sardinia and those circulating in Italy. A comparative analysis of these strains could provide insights into the viral transmission dynamics within the country.
Reviewer 2 Report
Comments and Suggestions for Authors
The manuscript represents a valuable contribution to the molecular epidemiology of BVDV by providing, for the first time, the whole-genome sequences of ten strains from Sardinia, Italy. The central finding-that eight of the ten strains form a distinct, new cluster within the BVDV-1b subtype-is interesting and has potential epidemiological significance. The use of WGS is a plus, enabling a highresolution view surpassing single-gene-based studies, for example 5'UTR. However, the manuscript in its current form has significant weaknesses in data presentation, methodological description, statistical support, and the interpretation of results that need to be addressed before publication.
Major
Title and "Preliminary" Nature
The title correctly describes the work as "Preliminary." That is honest, but it minimizes the impact. The data (10 full genomes from a defined area over 6 years) are substantial for a regional study. The authors should consider a more confident title, such as "Identification and Characterization of a Novel BVDV-1b Cluster in Sardinia through Whole Genome Sequencing," and remove the "preliminary" modifier from the text. The conclusions, while needing refinement (see below), are more conclusive than "preliminary" implies.
Quality of assembly and influence on analysis: This is the biggest technical flaw in the manuscript. Table 3 shows alarmingly high values for # N's per 100kbp, with a minimum of ~374 and a maximum of ~6748. On a ~12.3 kb virus genome, a value of 2027.89, like that in PV871979, would mean that there were around 250 unknown bases on the final assembly, which seriously compromises all downstream analysis.)
This has several implications: Variant analysis (Table 4, Figure 3) is unreliable. A high number of Ns will lead to underestimating the true number of variants. Snippy pipeline will likely treat these regions as un-informative. Reported numbers of SNPs in this context are questionable (721-950).
The alignment on which the phylogenetic inference is based allows for up to ~5% of data for some samples to be missing or ambiguous. This can distort branch lengths and topological accuracy.
The claim for a "Sardinian core genome" supported by 61 shared variants is weakened because such "shared" gaps or missing data could be artifacts of poor assembly in certain genomic regions common to the eight strains.
Authors must address this directly. They should provide the total number of Ns per genome, and discuss how gaps were considered in the multiple sequence alignment, for example if treated as missing data or gaps. Such a sensitivity analysis, running the phylogeny after excluding parts of the genome bearing a high quantity of missing data, would reinforce the results. Justification for the low assembly quality (for instance, high host contamination, low viral load) should be moved from the results to the methods/discussion.
Phylogenetic analysis and statistical support: The phylogenetic methodology is not fully described, and the results lack essential statistical information.
Model selection is not specified. It is good that the model used, GTR+G, is indicated, but how was it selected? Some tool like ModelTest-NG or similar should be used and mentioned. The sentence "Initial tree(s) for the heuristic search were obtained automatically." is directly copied from MEGA output and should be rephrased for a manuscript.
Results: The "100" bootstrap value mentioned was for either the whole tree or a specific node, but which one is unclear. The main assertion-that the eight Sardinian strains form a new niche-requires a high bootstrap value at the node defining that clade. This value is not provided. Similarly, support for the BVDV-1b clade and placement of PV871986/PV871987 require bootstrap values.
The phylogenetic tree (Figure 2) should show all relevant bootstrap values on the major nodes. The manuscript text should explicitly indicate the bootstrap value supporting the monophyly-that is, common ancestry-of the eight Sardinian strains. A value ≥70 is generally considered significant.
Overinterpretation of the "Core Genome" and variant impact: The term "core genome" is misused. In microbial genomics, the core genome is a set of genes shared by all members of a group. Here, the authors refer to 61 shared variants (SNPs/indels) relative to a reference, not a set of genes. They should use an expression that is more accurate, like "shared derived mutations" or "signature mutations." The discussion of the 19 missense mutations is highly speculative. While the functions of E2, Core, and NS2 are well described, the authors have provided no evidence that the specific mutations they identified, for example, p.Ser944Gly and p.Ala1140Ser, have any functional effect. Statements such as "could result in potential alterations," "could facilitate immune evasion," and "could influence viral protein production" are not supported by any in silico structural analysis or experimental data.
The Authors should change "core genome" to "set of shared characteristic variants" or similar.
Temper the language regarding the functional impact of the mutations. While it is fine to speculate that the mutations in key proteins could be important, this must be framed as a hypothesis for future research and not a finding of this study. A short in silico analysis using bioinformatics tools such as SIFT or PROVEAN to predict the deleteriousness of these mutations would greatly strengthen this section.
Data presentation and clarity: There are several confusing presentations of data in the manuscript.
Table 2: The "% Assembled" is very low (0.00119% to 0.97%). This shows host contamination is extreme, yet it is given without remark. It needs to be explained that this is expected in clinical samples without previous viral enrichment. Table 4 is mislabeled; its caption is the same as Table 3. The division into "Synonymous," "Missense," etc. is good, but without knowing the baseline variation in the broader BVDV-1b clade these totals are hard to interpret.
Minor and Suggestions
Abstract: "Phylogenetic niche" is a vague term. Consider "a distinct, well-supported monophyletic cluster."
The introduction is overall broad, but could be a bit more concise. The step from global to Italian to Sardinian context flows logically.
Methods - SISPA: The modifications made to the Marcacci et al. protocol should be briefly stated.
Results - Preprocessing: This sentence, "NovaSeq sequencing was used only for samples that performed poorly. with MiSeq", is out of place in the results; it belongs in Methods, in the Sampling/Sequencing Strategy section.
Discussion: The association of BVDV-1b with persistent infection/immunosuppression is stated without reference. Please provide a citation, for example, reference 26, Gallina et al., 2021.
Discussion: It is an overstatement in the conclusion that these findings will contribute to the development of intervention plans. The study identifies a local cluster but does not test vaccines or diagnostics against it. A more measured conclusion might be that it provides a genetic baseline for future studies.
Round 2
Reviewer 2 Report
Comments and Suggestions for Authors
The revised manuscript "Identification and Characterization of a Novel BVDV-1b Cluster in Sardinia through Whole Genome Sequencing" represents a substantial and excellent improvement over the previous version.
However, before final acceptance, I recommend a final round of minor revisions to polish the presentation, ensure consistency, and further sharpen the interpretation. The changes needed are primarily editorial.
Text Page 7: "The former segregated with strain EF101530 (DEU1) (100% bootstrap).". This is a bit ambiguous. Suggest: "The former formed a clade with strain EF101530 (DEU1) with 100% bootstrap support."
Discussion:
Paragraph 4, Page 10, starting "There are 25 complete Italian genomes.": This paragraph seems like a data dump and breaks up the flow of the narrative on the Sardinian cluster. It might be better to move this descriptive list of Italian strains to the Supplementary Materials and summarize the main point in the main text: e.g., "Notably, within the global dataset, the eight Sardinian strains described here form the largest and most recent monophyletic group of Italian BVDV-1b full genomes, highlighting the distinct and active circulation of this lineage.
Functional Predictions, Page 11: The text correctly points out low SIFT confidence scores. This could be reinforced briefly with an explanation of why this is the case, for example, "likely due to the limited diversity of BVDV sequences in the reference database used by the tool". This puts the "predictive" nature into context for the reader.
Reference formatting: Be constant - some journal names are italic, others not; some have dots after initials, others do not. This is often handled by the journal's production staff, but a consistent draft is helpful.
Comments on the Quality of English LanguageThe language is understandable but usually non-idiomatic, grammatically faulty, and at times awkward. The impression is often of a literal translation or writing by non-native speakers without after-the-fact professional editing. This distracts from the excellent science and could lead to misunderstandings or a perception of carelessness.
Here is a detailed assessment and recommendation:
Overall assessment: Needs professional editing in language
The manuscript is still not at the standard of polished, publication-ready English expected by journals such as Viruses.
Grammar and Syntax:
Article Usage: Incorrect or missing "a, " "an, " and "the.
Example (Intro, P2): "a globally distributed pathogen belonging to the Pestivirus genus of the Flaviviridae family" Correct. But later: "due to their genetic heterogeneity" should be "its" referring to "the genus".
Subject-verb agreement & tense:
Example (P2, Results): "The analysis revealed that despite the Sardinian strains' proximity they differed from it for a high number." → "differed from it by a high number." or "contained a high number."
Prepositions: Commonly misused.
"collected in EDTA" → "collected into EDTA tubes"
"differed from it for a high number" → "by a high number" or "in having a high number"
"implication in diagnostic and immunology" → "implications for diagnostics and immunology"
Awkward/Non-idiomatic phrasing:
Example (Abstract): "suggesting this may represent a specific Sardinian core genome." → Based on your fix, this is better as: "suggesting a distinct Sardinian genomic signature."
Example (P6, Results): "a result directly attributable to the absence of specific viral enrichment steps prior to sequencing." → This is clear but very stiff. Better: "a direct result of not using viral enrichment steps prior to sequencing."
Example (P10, Discussion): "The application of WGS without prior specific viral enrichment inevitably resulted in a predominance of host reads, reflecting the inherent challenge of sequencing directly from complex clinical matrices." → This is actually quite good, showing improvement. However, the next sentence is more awkward: "To mitigate the consequent low viral-to-host ratio.
Word choice errors:
Example (P2, Intro): "clove-hoofed animal species" → Correct expression: "clovenhoofed"
Example (P7, Fig 2 Caption): "The percentage of replicate trees in which the associated taxa clustered together." → That is standard phylogenetics language, so it is fine, but the passive voice is heavy.
Example (P11, Discussion): "were identified as robust markers" → In context, "reliable markers" or "strong candidates" might be better.
Punctuation and flow: Sentences are long and often comma-spliced, which makes them hard to follow.
Example (P4, Methods): "RNA was retrotranscribed into single strand cDNA (ss cDNA) using 200 units in the presence of the random primer FR26RV-N., 1X SSIV buffer, 10 mM dNTPs," → The list is cumbersome. Consider breaking into bullet points in the methods section for clarity, or using a more structured sentence.
There are also many typographical errors. In many cases, there are no spaces between words.
